# Thermal Effect on Thin-Film Formation of the Polymer Sheets by the CO_2_ Laser with the Copper Base

**DOI:** 10.3390/polym14173508

**Published:** 2022-08-26

**Authors:** Nobukazu Kameyama, Hiroki Yoshida

**Affiliations:** Department of Electrical, Electronic and Computer Engineering, Faculty of Engineering, Gifu University, 1-1 Yanagito, Gifu City 501-1193, Japan

**Keywords:** polymer, polystyrene, polypropylene, polyethylene terephthalate, laser processing, CO_2_ laser, thin film processing

## Abstract

A method that makes polymer sheets partially thinner with continuous-wave carbon dioxide (CO_2_) lasers has been developed. This method can create thin polymer films by attaching the polymer sheets to the copper base by vacuum suction through the holes in the base. Applying the method to polypropylene (PP), polyethylene terephthalate (PET), polystyrene (PS), and polytetrafluoroethylene (PTFE), the thin-film formation is confirmed in PP, PET, and PS but not PTFE. These polymers have the similar thermal properties. PP, PET, and PS show fluidity with increased temperature, but PTFE does not have fluidity. These characteristics of the polymers indicate that the fluidity of polymer is the important characteristic for film formation. The experiments with PP and PET sheets of different thickness show that thicker sheets make thicker films. The fluid flow of the molten polymer is considered to form the thin film at the bottom of the groove made by laser scribing. The numerical simulation of the 2D thermal model also indicates the week cooling effects of the base on the film formation and importance of polymer fluidity. The results of Fourier transform infrared spectrometer (FT-IR) show thermal degradation of the films. To decrease the heat’s effect on the films, the polymer sheets should be processed at the highest laser-beam scanning speed that can make thin films.

## 1. Introduction

Carbon dioxide (CO_2_) lasers have been widely used in laser processing in many industrial fields [1,2,3,4]. CO_2_ lasers have many advantages compared with other lasers. CO_2_ lasers can operate at the high power of 10 kW or more, which makes the machining time shorter, and are available at a reasonable cost because they do not use expensive materials. Some types of lasers, for example, diode lasers and fiber lasers, have become available at relatively low costs in recent years [4,5]. CO_2_ lasers are powerful equipment for laser processing because the far-infrared wavelength that almost all polymers absorb is unique. CO_2_ lasers are suitable for manufacturing inexpensive polymer products requiring a fast processing speed and low running costs for their operation [6].

There are several methods of making thin polymer films with lasers: laser chemical vapor deposition (CVD), pulsed laser deposition (PLD), and laser-induced forward transfer (LIFT). In laser CVD, deposition reactions are induced by focusing laser beams on the gaseous materials that are to be deposited on substrates, or laser beams are irradiated to substrates for the localized deposition of thin films [7]. The latter method can realize micro-patterning of the substrates since the reaction area is limited to the spot size of laser beams. PLD makes use of laser ablation, which is generated by focusing high-power laser beams to film-forming materials [8,9]. The material plumes caused by laser ablations are deposited to substrates, resulting in the formation of thin films. This method, however, is not adequate for polymer materials because the laser ablations can damage polymer materials. Matrix-assisted pulsed laser evaporation (MAPLE) was developed to prevent this [9,10,11,12,13,14]. The method uses a frozen solvent containing the materials to be deposited. The solvent is volatile and absorbs the laser energy rather than the materials. The materials are ejected by ablating the solvent with less damage than conventional PLD. PLD and MAPLE can deposit polymer films with a large area. A mask is used to make deposition patterns [15,16]. MAPLE is applied to a direct writing technique, called matrix-assisted pulsed laser evaporation direct writing (MAPLE DW) [17,18,19]. In the method, the solvent is coated to a transparent support. Laser beams are focused on the solvent through the support and the solvent is transferred to a receiver substrate by laser ablations. LIFT is schematically similar to MAPLE DW [8,20,21,22]. Instead of the solvent, the donor material to be transferred is coated to the support. The conventional LIFT also has the same kind of problem as PLD, where the laser ablations cause thermal decomposition of the donor material. To avoid any damage to the material, a sacrificial layer is added between the support and the donor material. Laser beams are irradiated to the layer and the ablations of the layer transfer the donor to the receiver. The method is called dynamic release layer (DRL) LIFT [22,23,24,25].

We developed a new method of thin-film processing of polypropylene (PP) and polystyrene (PS) sheets by a CO_2_ laser with a copper base [26]. In the conventional way of cutting polymer sheets, there is no supports or base on the backside of the processing area to prevent damage. It is difficult to make thin films on polymer sheets in the conventional way. Most thermoplastics, such as PP and PS, can be deformed by heat, since processing with continuous-wave (CW) CO_2_ lasers is thermal processing. Therefore, the heat-affected zone (HAZ) is not ignorable. In the new method that we developed, the backside of polymer sheets is attached to the copper base during laser-processing by vacuum suction through holes in the base, which makes it possible to form 50–100 µm-thick films on the polymer sheets. It is confirmed that aluminum, brass, and iron are also available for this method. Aluminum and brass have similar physical properties to copper, that is, a low absorption of CO_2_ laser beams, high heat conductivity, and higher melting points than polymer decomposition temperatures. Iron absorbs around ten percent of the beams’ energy [27], which is ten times more than copper. Films of the same thickness are formed with all bases.

The films formed by this method are thicker than the films transferred by MAPLE [28], or DRL LIFT [20,25]. MAPLE and DRL LIFT can control film thickness with a precision of one micron or better. Therefore, it takes time to make a film with a thickness of several ten micrometers. This method can process at a faster speed with fewer processing steps, although it is difficult to control film thickness with high precision.

In this study, we investigate the thermal effects of the thin-film processing method on the polymer sheets. The method is applied to the sheets of PP, polyethylene terephthalate (PET), PS, and polytetrafluoroethylene (PTFE) to survey the dependency of this method on the polymer characteristics. The mechanism of thin-film formation is discussed, looking at the polymers’ physical properties and the processing characteristics of the polymers. To confirm the thermal effects of the copper base on the polymer sheets, a two-dimensional thermal simulation is performed. The thermal influence on the sheets is analyzed with Fourier transform infrared spectrometer (FT-IR).

## 2. Materials and Methods

The experiments were performed with the 10.6 µm CW CO_2_ laser (Videojet 3330, Videojet X-Rite K.K., Tokyo, Japan). The beam was controlled with Galvo mirror system, the beam size at the focal point was around 300 µm in diameter, and the focal length was 150 mm. The laser power *P* was fixed at 30 W and the scanning speed *S* was controlled from 1 to 500 mm/s. The polymer sheet materials used in this study were transparent PP, transparent PET, white translucent PS, and white PTFE. The thermal and optical characteristics of the polymers are summarized in Table 1. The specific heats, glass transition points, and melting points were measured by differential scanning calorimetry (DSC) (DSC 2500, TA Instruments, New Castle, DE, USA). Decomposition temperature was measured by simultaneous DSC and thermogravimetric analysis (SDT) (SDT Q600, 2500, TA Instruments, New Castle, DE, USA) under nitrogen atmosphere. Absorption coefficient at the wavelength of 10.6 µm was measured by FT-IR (FT-IR4700, JASCO, Tokyo, Japan). Melt viscosities were measured by rheometer (AR-G2 KG, TA Instruments, New Castle, DE, USA).

The base has four holes with a diameter of 2.0 mm. The polymer sheets were attached to the base by vacuum suction through the holes with a vacuum pump (LV-125A, NITTO KOHKI, Tokyo, Japan) during laser processing. The beam was focused on the surface of the polymer sheet and scanned linearly once with no assistant gas at room temperature (RT). The schematic of laser processing with the copper base is shown in Ref. [26].

## 3. Results and Discussions

### 3.1. Thin-Film Processing of PP, PET, PS, and PTFE

The results of thin-film processing of PP, PET, PS, and PTFE sheets were compared with those of the conventional method, which does not use any base under the area processed by the laser beams, i.e., the backside of the polymer is in air. The polymer sheets with thickness of around 300 µm are used. 

Figure 1 shows the cross-section of a PP sample processed under conditions of 167 J/cm^2^. The thickness at the bottom of the groove and width of the groove at the height of the initial surface were measured. The samples were covered with silicone putty after processing to avoid deformation by cutting.

Figure 2 shows the thickness results of thin-film processing of the polymer sheets. In the absence of the base, the thicknesses of all polymer sheets decrease as the laser energy increases, and the sheets are finally cut at around one hundred J/cm^2^. The film formation was confirmed in PP, PET, and PS with the copper base. In PP, the decrease in the thickness stopped at ~170 J/cm^2^ and the formed films were around 70 µm in thickness. In PET, the formation of an around 50 µm-thick film was confirmed over ~130 J/cm^2^. Holes in the film were observed at 500 J/cm^2^. The method is also applicable to PS, but the energy range of film formation is narrower than in PP and PET. The film thickness of PS was around 100 µm. Cracks were observed in the film at 167 J/cm^2^. Only PTFE sheets were cut, regardless of whether the base was used.

At first, cooling effect of the base on the polymer sheets is discussed. Copper has several thousand times higher heat conductivity (~400 W/m·K) than the polymers. The copper base is hardly heated by CO_2_ laser beams, since the absorptance is one percent or less [30]. Therefore, the base can cool the polymer sheet effectively. As shown in Table 1, the heat conductivities of all used polymers have almost the same values. The cooling effects of the base on PTFE should be same as the other polymers, because the same base is used, the PTFE sheets are attached to the base in the same way, and heat transfer from the polymers to the base does not depend on materials. Thin-film formation was also confirmed with an iron base [26]; nevertheless, the iron base has low heat conductivity (52 W/m·K) and high absorptance (~10%) of CO_2_ laser beams [27]. The films formed with copper or iron bases, however, have almost the same thickness, even though the temperature of the iron base should be higher than that of the cooper base. If the backside of the polymer sheets is in thermal equilibrium with the bases and the thin film is formed, the film thickness should be influenced by the base temperature. It is presumed that the cooling effect of the base will not be significant.

Secondly, almost all thermoplastics, such as PP, PET, and PS, show fluidity with increased temperature and solidify upon cooling. PTFE, however, has quite a high viscosity at a temperature above the melting point, although it is also thermoplastic.

One hypothesized mechanism of film formation is that the fluid flow of the molten polymer to the bottom of the groove forms the thin film by laser-scribing. Heated by laser irradiation, the polymer sheet melts and a part of it decomposes to gas. Molten polymer, which can deform by gravity but is not heated to the point of decomposition, flows to the bottom of the groove during and/or after laser scribing before losing fluidity by cooling. In this case, the viscosity of molten polymer, which is called melt viscosity, is an important parameter for this phenomenon. Figure 3 shows the temperature dependence of the melt viscosity of PP, PET, and PS, as measured by rheometer. The measurement temperature range is from the temperature at which the polymers show fluidity to the temperature at which thermal decomposition starts. The melt viscosity of PP and PET falls below 10 Pa·s and PS shows a higher melt viscosity than the others. PP and PET, showing high fluidity, are examined in terms of the volume of the molten polymer in the following section.

### 3.2. Relationship between the Sample Thickness and the Formed Film Thickness

In order to confirm the effect of polymer thickness on the thin-film formation, the experiments were conducted with PP sheets with thicknesses of 0.24 and 0.50 mm, and with PET sheets with thicknesses of 0.20 mm and 0.53 mm. The experimental conditions were the same as those mentioned above.

Figure 4 and Figure 5 show the experimental results with PP and PET sheets of different thickness. Both polymers show the dependence of film thickness on sheet thickness, that is, thicker sheets form thicker films. The films that were formed should have the same thickness regardless of the sheet thicknesses if the backside of the polymer sheets is in thermal equilibrium with the base. The volume of molten polymer increased as the sheets grew thicker, but the groove widths did not change according to the sheet thicknesses, resulting in an increase in the film thickness. The results support the hypothesis that films are formed by the fluid flow of the molten polymer.

### 3.3. Numerical Simulation of the Thermal Model and Pseudo-Fluid Model

The thermal model in a two-dimensional axisymmetric configuration, which does not include hydrodynamic effects, was constructed to estimate the thermal effect of the base on the polymer in PP and PET. The simulation schematic is shown in Figure 6. The calculation size of x-direction was five times larger than the beam radius of 150 µm. The 300 µm thick polymer sheet was attached to the copper base in air atmosphere and there was no gap between the sheet and base. The laser beam of two-dimensional Gaussian distribution scanned the surface of the polymer sheet in the y-direction. No reflection was considered on the polymer surface. The beam was absorbed by the polymer based on Lambert–Beer’s law. The copper base reflected all beam transmitted through the polymer. The reflected beam was absorbed by the polymer again. The heat caused by the absorption of the laser beam propagated in the polymer in x-z plane and was transferred from the polymer surface to air and from the polymer backside to the copper base. Air, the base, and the polymer lateral boundary remained at a constant temperature of 20 degrees. The liquid phase of the polymer was not treated and the latent heat of melting was included in the specific heat to obtain the total energy necessary to heat this to the decomposition temperature. The volumes heated to the decomposition temperature and obtaining the energy over the latent heat of decomposition were replaced by air, which does not disturb the laser beams. The heat transfer coefficient of free convection of air was 5 W/(m^2^·K). Heat transfer coefficient between the polymer and the copper base was 1,500 W/(m^2^·K) in Ref. [31]. The physical property values used in the simulation are shown in Table 2. No values changed with temperature.

The simulation results of the thermal model of PP and PET at 167 J/cm^2^ are represented in Figure 7. The figures show the temperature distribution in the polymers. The polymers start to decompose at the center of laser irradiation and are replaced by air, which is described as the blue areas. The thicknesses at the groove bottom after the laser beam passes are measured. At the condition, PP at the boundary with the base remains without penetrating, which is attributed to the cooling effects of the base, but the thickness of the film is thinner than the thickness of the experiment, while PP is completely penetrated.

In the actual phenomena of laser processing, the molten polymer simultaneously flows at the same time as laser-scribing due to gravity and recoil pressure, which affects the groove shape. In order to simplify the phenomena, the volume of the molten polymer whose viscosity is lower than 100 Pa·s is averaged to the groove bottom after calculation of the thermal model instead of incorporating hydrodynamic effects. This is called the pseudo-fluid model in this paper.

Figure 8 shows a comparison of the thermal and pseudo-fluid models with the experimental results. The thicknesses of the thermal model rapidly decrease as the laser energy increases and becomes zero in both polymers, as shown by the blue triangles. The cooling effects of the base are lower than the heat caused by laser absorption even though the cooling effects are overestimated because the absorption of the laser beam caused by the base and heat transfer from the polymers to the base are not considered. In contrast, the pseudo-fluid model qualitatively shows a similar trend to the experimental results, as shown by the green squares. These results indicate that there are few cooling effects of the copper base on the polymer sheets and show the importance of polymer fluidity. The reason that the simulation results do not fit with the experimental results at low energy levels is considered to be that polymers gradually start decomposing with increasing temperature and the decomposition temperatures used in the simulations are at the maximum heat-flow points.

### 3.4. FT-IR Analysis of the Formed Thin Films

The formed films are considered to be degraded by heat since the molten polymers have high fluidity at temperatures near sublimation points. In order to confirm this, the samples with a thin-film area of 10 × 5 mm^2^ were prepared by several laser irradiations with a scanning interval *d*. The length of one scan was 10 mm. The processed surface of the films was measured with attenuated total reflection (ATR) FT-IR. The infrared spectra are shown in Figure 9. Figure 9a(1)–(3) represent the infrared spectra of PP at *d* = 0.6 mm and *S* = 5 mm/s, *d* = 0.4 mm and *S* = 100 mm/s, and the original sample, respectively. All results show the almost same spectra, but only the original sample does not have a peak at 1650 cm^−1^, as shown in the inlet of Figure 9a. The peak in the processed samples shows the existence of C=C bonds [33]. The processed samples have another peak at 886 cm^−1^, which originated from vinylidene =C−H out-of-plane bend [34]. Carbon double bonds in polyolefin are caused by thermal degradation [31]. Figure 9b(1)–(3) represent the infrared spectra of PET at *d* = 0.4 mm and *S* = 40 mm/s, *d* = 0.4 mm and *S* = 100 mm/s, and the original sample, respectively. PET has typical peaks in carbonyl C=O stretching (1712 cm^−1^), C–O stretching of ester group (1236 and 1089 cm^−1^), and the out-of-plane deformation of the two carbonyl substituents on the aromatic ring (722 cm^−1^) [33,34]. These peaks in the processed sample at *S* = 100 mm/s, as shown in Figure 9b(2), slightly decrease compared with the original sample, while the spectrum of the sample at *S* = 40 mm/s shown in Figure 9b(3) completely changed. At the first stage of PET’s thermal degradation, the scission of carboxyl and vinyl ester groups occurs and yields COOH on the aromatic ring. At the next stage, decarboxylation occurs, releasing CO_2_ from COOH. The simple aromatic rings that remain react with each other, leading to cross-linking [34]. The carbonyl band, ester band and carbonyl substituents on the aromatic ring are removed by thermal degradation, resulting in decreases in the peaks, as shown in Figure 9b(3). Figure 9c(1)–(3) represent the infrared spectra of PS at *d* = 0.4 mm and *S* = 75 mm/s, *d* = 0.4 mm and *S* = 100 mm/s, and original sample, respectively. All PS results also show almost the same spectra. Butadiene was reacted with PS in order to overcome the brittleness of crystal polystyrene [35]. The composite is called high-impact PS with a translucent or opaque color. The peak at 965 cm^−1^, as shown in the inlet of Figure 9c, represents the trans C−H out-of-plane bending of butadiene [33]. The processed sample has a lower peak than the original one, which represents decomposition of butadiene [31].

Thermal influence is inevitable in this method due to the thermal processing with CO_2_ lasers. All processed samples, therefore, are influenced by heat. The heat degradation of PET at a low scanning speed is remarkable, since it is exposed to heat for a long time. To decrease thermal influence, the polymer sheets should be processed at the highest scanning speed that can form thin films.

## 4. Conclusions

It is confirmed that the thin-film processing method with the copper base is suitable for PP, PET, and PS but not PTFE. These polymers have similar thermal characteristics, while PP, PET, and PS show high fluidity at a high temperature, but only PTFE has quite a high viscosity above the melting point. The fluidity of polymer is considered to be important for film formation and molten polymer flows to the bottom of the groove. The experiments with PP and PET sheets of different thicknesses show that thicker sheets form thicker films. The films are considered to get thicker, since the thick sheets have a more molten volume than thin ones and the groove widths do not change with the sheet thicknesses.

The thermal model including heat conductivity and heat transfer is simulated to survey the cooling effects of the base on the polymer sheets. The simulation results of the thermal model shows that the copper base does not have enough cooling effects on the polymer sheets for form formation. After model calculation, the volumes of molten polymer are averaged to the bottom of the groove by laser-scribing, which is called the pseudo-fluid model. The pseudo-fluid model shows a similar tendency to the experimental results. The flows of molten polymers, however, occurred during laser irradiation by gravity and recoil pressure by decomposition or evaporation.

The spectra of ATR FT-IR show the heat degradation of the thin films. The spectrum of PET processed at slow scanning speeds greatly changed due to heat. To decrease the influence of heat on the films, the laser beams should be scanned at the highest speed that can create thin films.

## Figures and Tables

**Figure 1 polymers-14-03508-f001:**
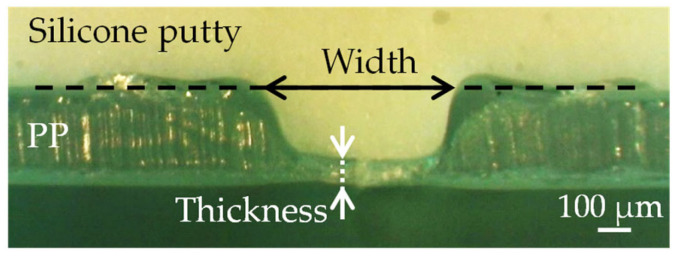
Cross-section of a PP sample processed in the condition of 167 J/cm^2^.

**Figure 2 polymers-14-03508-f002:**
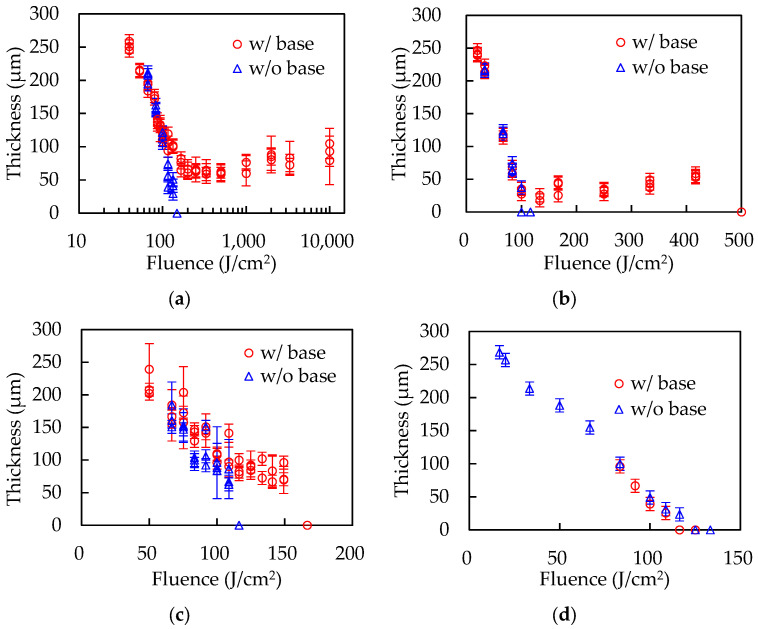
The thickness results of (**a**) PP, (**b**) PET, (**c**) PS, and (**d**) PTFE with or without the base.

**Figure 3 polymers-14-03508-f003:**
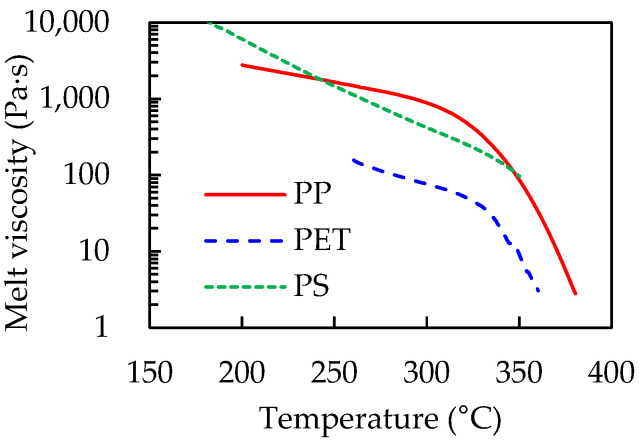
Temperature dependence of melt viscosity of PP, PET, and PS.

**Figure 4 polymers-14-03508-f004:**
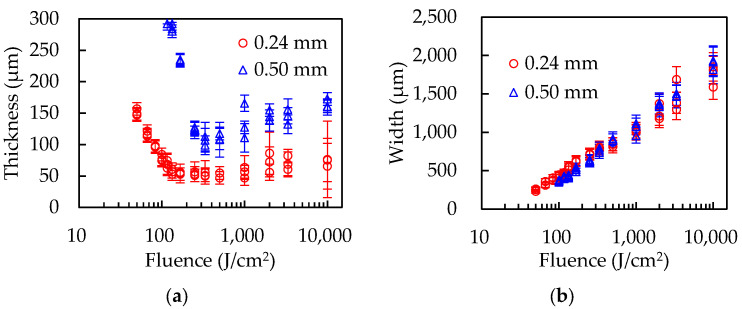
The results of (**a**) film thicknesses and (**b**) groove widths with the PP sheets 0.24 mm and 0.50 mm thick.

**Figure 5 polymers-14-03508-f005:**
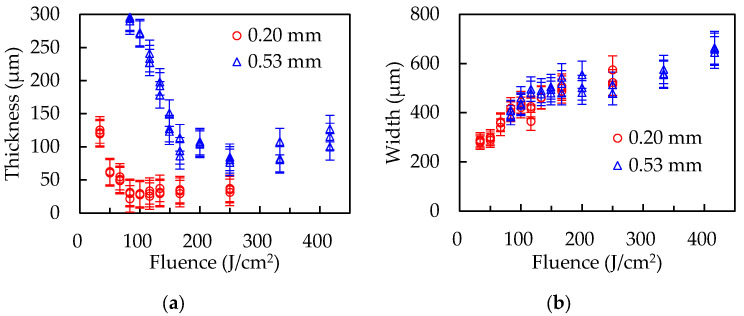
The results of (**a**) film thicknesses and (**b**) groove widths with the PET sheets 0.20 mm and 0.53 mm thick.

**Figure 6 polymers-14-03508-f006:**
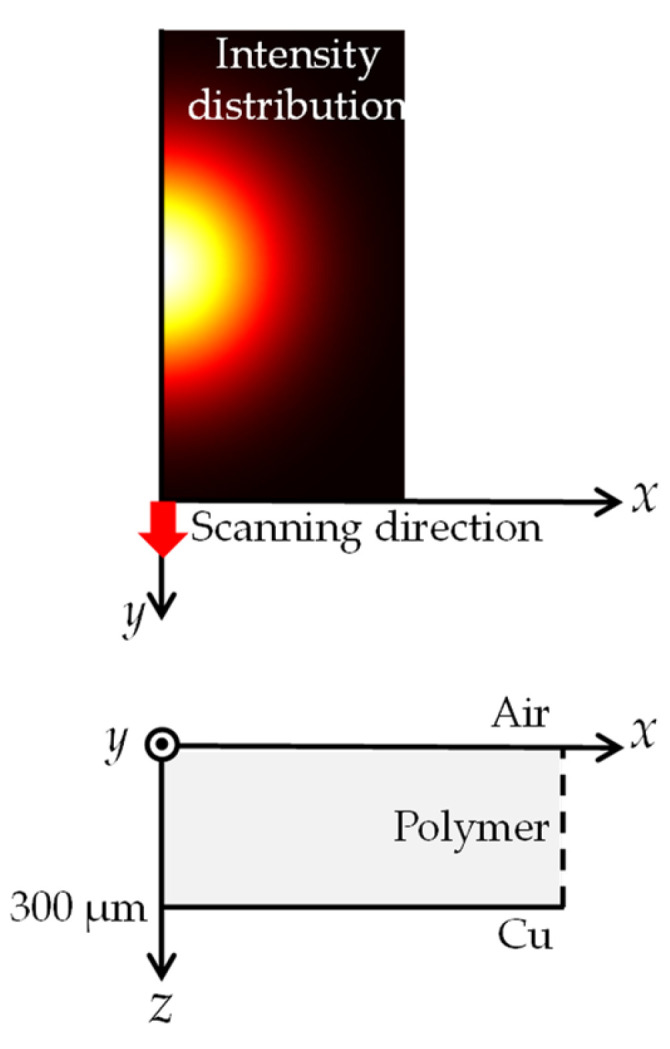
Simulation schematic of the laser-scanning direction and the calculation area.

**Figure 7 polymers-14-03508-f007:**
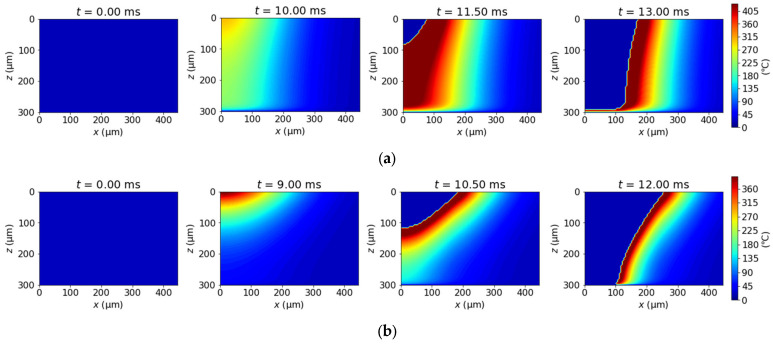
The simulation results of the thermal model of (**a**) PP and (**b**) PET at 167 J/cm^2^. The laser center at t = 0.00 ms is 750 µm apart from x-z plane.

**Figure 8 polymers-14-03508-f008:**
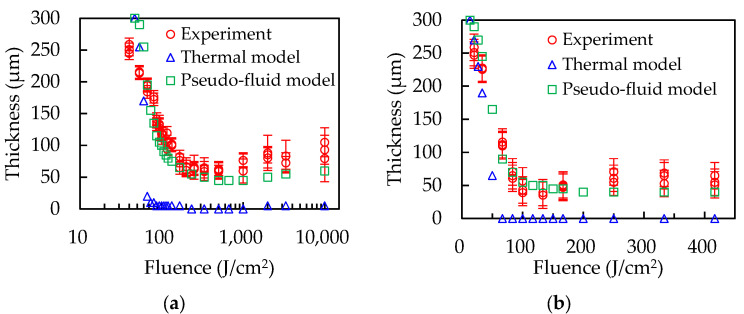
Comparison of the thermal and pseudo-fluid model with the experimental results. (**a**) PP, (**b**) PET.

**Figure 9 polymers-14-03508-f009:**
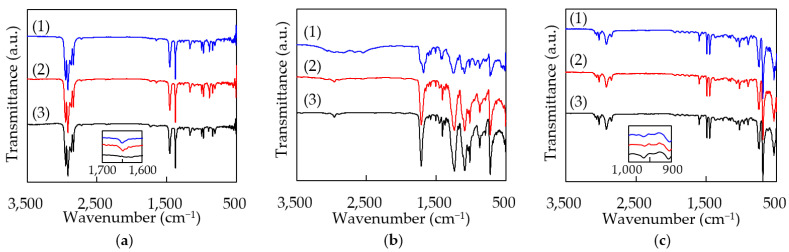
The infrared spectra of the processed surfaces of the films. (**a**) PP, (1) *d* = 0.6 mm, *S* = 5 mm/s, (2) *d* = 0.4 mm, *S* = 100 mm/s, (3) original, (**b**) PET, (1) *d* = 0.4 mm, *S* = 40 mm/s, (2) *d* = 0.4 mm, *S* = 100 mm/s, (3) original, (**c**) PS, (1) *d* = 0.4 mm, *S* = 75 mm/s, (2) *d* = 0.4 mm, *S* = 100 mm/s, (3) original.

**Table 1 polymers-14-03508-t001:** Thermal and optical property values of the polymers.

	Specific Heat (J/g·°C) of 25 °C	Thermal Conductivity (W/m·K) *	Glass Transition Point (°C)	Melting Point (°C)	Decomposition Temperature (°C)	Absorption Coefficient (cm^−1^) of 10.6 µm	Melt Viscosity (Pa·s)
PP	1.46	0.12	−3^*^	164	459	30	10^3^ (290 °C)
PET	1.37	0.15	67	250	420	110	10^2^ (280 °C)
PS	1.51	0.14	97	-	413	93	10^3^ (270 °C)
PTFE	1.05 *	0.25	31 *	327 *	540 *	50	10^10^ (380 °C) *

* Ref. [29].

**Table 2 polymers-14-03508-t002:** The physical property values used in the simulation.

	Specific Heat (J/g·°C)	Density (g/cm^3^) *	Latent Heat of Decomposition (J/g)
PP	2.77	0.90	378
PET	2.00	1.37	184

* Ref. [32].

## Data Availability

The data presented in this study are available on request from the corresponding author.

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
