# Peer review of "Thermal Effect on Thin-Film Formation of the Polymer Sheets by the CO2 Laser with the Copper Base"

_polymers, 2022, doi:10.3390/polym14173508_

Round 1

Reviewer 1 Report (Previous Reviewer 1)

The mentioned comments have been addressed. The manuscript can be prepared for publishing.

Thank you.

Author Response

Thank you for your review. English language in the manuscript was edited.

Reviewer 2 Report (Previous Reviewer 2)

Dear Authors

Your manuscript is an improved version of the previous one. There is still possibility to be improved further in terms of including novel references as well as English editing. Abstract also needs more refining more clear sentences without repetition. The list of keywords should be extended with specific ones.

Suggestions are also given in the attached version of your article.

Thanks.

Author Response

Some references and keywords were added.

Abstract was revised.

This manuscript is a resubmission of an earlier submission. The following is a list of the peer review reports and author responses from that submission.

Round 1

Reviewer 1 Report

The authors have prepared an original manuscript. It undertakes the study of polypropylene, polystyrene, polyethylene tetraphytalate, polytetrafloroetylene in interaction with a copper substrate which results in various thermal effects. The evaluation of these interactions were made by a two dimensionel thermal simulation as well as experimental work based on having COlaser beam exposed to polymer coated copper substrates. The points to be amended are below:

(1) The research design is appropriate but there are very few references in the intorduction section to support the background of the work. The first paragraph of introduction should be revised by adding related references to the explanations.

(2) The results and discussion section also lacks sufficient reference citations. It would be contributory for the manuscript to include comparisons of the results obtained from the mentioned CO2 laser beam technique with more conventional techniques such as CVD and PLD.

Reviewer 2 Report

Dear Authors

Your manuscript is interesting and could be very important for industrial applications. However, the manuscript is not well written. English should be improved and a better explanation of the topic is required. Particularly, the first 3 section need further rewriting and improvement, Introduction part should be updated in order to reflect the up-to-date novelty, applicative notes and state of the art in the area of description in the article. The reference list is relatively old and not updated as well. Please find the attached copy with suggestions for improvement.
